# LABEL-EFFICIENT AUDIO CLASSIFICATION THROUGH MULTITASK LEARNING AND SELF-SUPERVISION

**Tyler Lee,**[⋆] **Ting Gong,**[⋆] **Suchismita Padhy,**[⋆] **& Anthony Ndirango**[⋆]
Intel AI Lab
Santa Clara, CA
`{tyler.p.lee,ting.gong,suchismita.padhy,anthony.ndirango}@intel.com`

**Andrew Rouditchenko**
MIT
Cambridge, MA
`roudi@mit.edu`

## ABSTRACT

While deep learning has been incredibly successful in modeling tasks with large, carefully curated labeled datasets, its application to problems with limited labeled data remains a challenge. The aim of the present work is to improve the label efficiency of large neural networks operating on audio data through a combination of multitask learning and self-supervised learning on unlabeled data. We trained an end-to-end audio feature extractor based on WaveNet that feeds into simple, yet versatile task-specific neural networks. We describe several easily implemented self-supervised learning tasks that can operate on any large, unlabeled audio corpus. We demonstrate that, in scenarios with limited labeled training data, one can significantly improve the performance of three different supervised classification tasks individually by up to 6% through simultaneous training with these additional self-supervised tasks. We also show that incorporating data augmentation into our multitask setting leads to even further gains in performance.

## 1 INTRODUCTION

Deep neural networks (DNNs) are the bedrock of state-of-the-art approaches to modeling and classifying auditory data (Amodei et al. (2015); van den Oord et al. (2016); Li et al. (2017)). However, these data-hungry neural architectures are not always matched to the available training resources, and the creation of large-scale corpora of audio training data is costly and time-consuming. This problem is exacerbated when training directly on the acoustic waveform, where input is high-dimensional and noisy. While labeled datasets are quite scarce, we have access to virtually infinite sources of unlabeled data, which makes effective unsupervised learning an enticing research direction. Here we aim to develop a technique that enables models to generalize better by incorporating auxiliary self-supervised auditory tasks during model training (Doersch & Zisserman (2017)).

Our main contributions in this paper are twofold: the successful identification of appropriate self-supervised audio-related tasks and the demonstration that they can be trained jointly with supervised tasks in order to significantly improve performance. We also show how to use WaveNet as a general feature extractor capable of providing rich audio representations using raw waveform data as input. We hypothesize that by learning multi-scale hierarchical representations from raw audio, WaveNet-based models are capable of adapting to subtle variations within tasks in an efficient and robust manner. We explore this framework on three supervised classification tasks - audio tagging, speaker identification and speech command recognition - and demonstrate that one can leverage unlabeled data to improve performance on each task. We further show that these results pair well with more common data augmentation techniques, and that our proposed self-supervised tasks can also be used as a pre-training stage to provide performance improvements through transfer learning.

---

[⋆]These authors contributed equally to this work.

## 2 RELATED WORK

Prevailing wisdom suggests that a single model can only learn multiple tasks if they are related in some way with some underlying structure common to them (Caruana (1997)). Such structure has been described for decades in the literature on sensory environments, with Gabor filters and gammatone filters underlying much of visual and auditory processing, respectively (Olshausen & Field (1996)). Perhaps models trained to accomplish many tasks might be able to synergize to uncover this underlying structure, enabling better single-task performance with smaller amounts of data per-task. We follow a relatively common approach to multitask learning aimed at learning a single non-trivial general-purpose representation (Bilen & Vedaldi (2017)). Examples of other intriguing approaches can be found in (Meyerson & Miikkulainen (2017); Ruder (2017)).

Much as shared representations allow models to pool data from different datasets, the problem persists that the cleanly labeled datasets that have permitted numerous breakthroughs in deep learning are painstaking to come by. One promising solution to label scarcity uses self-supervised learning to take advantage of unlabeled data. Self-supervised learning has shown promising results in the visual domain , leveraging unlabeled data using tasks like inpainting for image completion (Noroozi et al. (2017); Pathak et al. (2016b)), image colorization (Larsson et al. (2017); Zhang et al. (2016)), and motion segmentation (Pathak et al. (2016a)). Despite these efforts, little previous work has taken advantage of self-supervision in the audio domain.

## 3 EXPERIMENTAL SETUP

We implemented an end-to-end audio processing network that finds a common embedding of the acoustic waveform within a "trunk" network modeled after the WaveNet architecture (van den Oord et al. (2016)). The embedding is then processed by simple, independent, task-specific "head" networks. The trunk and head networks are trained jointly for each experiment described below. Our experiments consist primarily of models in which a single supervised "main" task is trained jointly with 0 to 3 self-supervised "auxiliary" tasks.

### 3.1 WAVENET TRUNK

Briefly (see appendix for details), our WaveNet trunk consists of 3 blocks of 6 dilation stacks each. Each dilation stack is comprised of a gate and filter module, with 64 convolutional units per module. The outputs from the filter and gate modules are (elementwise) multiplied and then summed with the input to the stack. These choices yield a WaveNet trunk with an effective receptive field length of $1 + 3(2^6 - 1) = 190$ samples or approximately 12 ms.

### 3.2 SUPERVISED TASKS

We tested our setup on three distinct supervised tasks: audio tagging, speaker identification and speech command recognition. Each is trained using a separate labeled dataset along with up to three self-supervised tasks trained with unlabeled data. Our description of the tasks is necessarily brief, with details relegated to the appendix.

The audio tagging task is trained on the FSDKaggle2018 (Gemmeke et al. (2017)) dataset collected through Freesound. This dataset contains a total of 11,073 files provided as uncompressed PCM 16 bit, 44.1 kHz, monaural audio which is further subdivided into a training set and a test set. Before being fed to the network, each audio segment is first cropped to 2 seconds and padded with zeros if the source clip is too short. Since the WaveNet trunk produces embeddings with a temporal structure, this task averages the output across time to produce a single output vector for the entire audio sequence, which in turn feeds into a single fully-connected layer with 512 units and ReLU nonlinearity, followed by a softmax output layer. Training is done by minimizing the cross entropy between the softmax outputs and one-hot encoded classification labels.

The speaker identification task is trained on the VoxCeleb-1 dataset (Nagrani et al. (2017)) which has 336 hours of data from 1251 speakers. Individual clips are sourced from interviews with celebrities in a variety of different settings. Data from each individual is sourced from multiple interviews and one interview is held-out to produce a test set with 15 hours of data. Before being fed to the network,

each audio segment is first cropped to 2 seconds in duration. Given the large variations in the audio quality of the samples in this dataset, we found it necessary to also normalize the clips and apply a pre-emphasis filter. This task's head architecture features a global average pooling layer, followed by 2-layer perceptron with 1024 units per layer, batch normalization and a ReLU nonlinearity. The output is then passed to a softmax layer and evaluated using a cross-entropy loss.

The speech command recognition task is trained on the Speech Commands dataset (Warden (2018)). The entire dataset consists of 65,000 utterances of 30 short words, formatted in one-second WAVE format files. There is a total of 12 categories; 10 words (yes, no, up, down, left, right, on, off, stop, go), with the rest classified as either unknown or silence. The speech command recognition head is a stack of three 1D convolutions. Between each convolutional layer we used batch normalization and dropout, followed by a ReLU nonlinearity. The three convolution layers have widths of 100, 50, and 25 and strides of 16, 8, and 4, respectively. The output is passed to a final softmax layer and evaluated using a cross-entropy loss.

### 3.3 SELF-SUPERVISED TASKS

We selected next-step prediction, noise reduction, and upsampling for our self-supervised, auxiliary tasks. They are easily implemented and can be synergistically paired with our main (supervised) tasks. The self-supervised tasks were trained on both the main task's data and unlabeled data sampled from the 100-hour and 500-hour versions of the Librispeech dataset (Panayotov et al. (2015)). This dataset was only used to train the auxiliary tasks.

All three auxiliary tasks share the same basic head architecture. They begin with two convolutional layers with 128 filters and ReLU nonlinearities and a final linear convolutional layer with 1 output unit feeding into a regression-type loss function (see appendix for details).

## 4 RESULTS

Our primary goal was to develop a multitask framework which is completely generic for audio, making it prudent to work with waveform inputs as opposed to, say, "high level" feature representations like spectrograms. While convolutional architectures trained on spectral/cepstral representations of audio can indeed give better classification performance than models trained directly on raw waveforms, they significantly restrict the range of audio processing tasks which they can perform. Thus, state-of-the-art baseline models for different tasks may vary wildly in their network architectures, subsequently limiting the amount of information that can be gained from a smaller pool of potential self-supervised tasks. If the goal is to understand the interaction between the learning dynamics of disparate tasks, then the focus should be on models which make the fewest assumptions about the representation of inputs. As such, we emphasize improvements in performance afforded by multitask learning relative to a single task baseline trained on raw audio. Closing the performance gap between models trained using spectral representations (e.g. Lederle & Wilhelm (2018); Nagrani et al. (2017)) and those trained on waveforms is left to future work.

### 4.1 MULTITASK LEARNING IMPROVES LABEL EFFICIENCY

Joint training with three self-supervised tasks proved beneficial for each of our three supervised tasks (Table 1). For the audio tagging task, multitask training improved MAP@3 score by .019 and top-1 classification rate by 1.62%, simply by including additional unsupervised tasks without increasing training data. Since the auxiliary tasks can be trained with unlabeled data, we gradually incorporated larger versions of Librispeech into our training regimen to investigate the effects of self-supervision.

Table 1: Multitask performance gains on supervised tasks w/ increasing amounts of unlabeled data.

|  | Audio Tagging | | Speaker ID | | Speech Command |
|---|---|---|---|---|---|
|  | MAP@3 Score | Top-1 (%) | Top-5 (%) | Top-1 (%) | Top-1 (%) |
| none (0) | 0.656 | 56.93 | 74.85 | 56.77 | 93.09 |
| train-clean-100 (100) | 0.671 | 58.39 | 74.82 | 57.34 | 93.39 |
| train-other-500 (500) | **0.693** | **61.39** | **75.22** | **57.61** | **93.78** |
| Baseline | 0.637 | 55.31 | 73.81 | 56.27 | 93.05 |

With each increase in unlabeled dataset size, we saw a further improvement on both performance metrics, with a MAP@3 increase of up to .056 with an additional 500 hours of unlabeled data. Using the same setup, but swapping the audio tagging task with either the speech command classification or the speaker identification task showed a similar, though more measured, trend with increasing amounts of unlabeled data. Speech command classification went from 93.05% in the baseline model to 93.78% when trained with an additional 500 hours of unlabeled data. Speaker identification on the VoxCeleb dataset was a much more challenging task for the network overall. There, top-5 classification performance peaked at 75.22%, up from the baseline performance of 73.81%.

## 4.2 MULTITASK LEARNING IS ADDITIVE WITH DATA AUGMENTATION

The results above show that multitask learning can improve the performance of any of our supervised tasks without any additional labeled data. To get an idea of the significance of the observed effects, we decided to compare the results above with another common technique for improving label efficiency: data augmentation. We trained a single task model on audio tagging with two different kinds of data augmentation: pitch shifting and additive noise (with SNRs of 10 to 15 dB).

Table 2: Audio tagging task due with data augmentation. NI=noise injection; PS=pitch shifting. MTL100=multitask learning with all auxiliary tasks trained on 100hrs of unlabeled data.

|            | MAP@3 Score | Top-1(%) |
| ---------- | ----------- | -------- |
| NI         | 0.661       | 57.31    |
| PS         | 0.703       | 62.60    |
| PS + MTL100 | **0.726**  | **64.87** |

We found that pitch-shift augmentation produced an increase in MAP@3 of .066, comparable to our largest multitask benefits (Table 2). Noise augmentation showed a somewhat smaller MAP@3 increase of .024. Interestingly, the performance gains from augmenting with noisy data are similar to those obtained by training the main task jointly with a self-supervised noise-reduction task. Finally, training with both pitch-shift augmentation and additional self-supervised tasks yielded a MAP@3 increase of .089 – our highest performance from any experiment – suggesting that both methods for improving label efficiency are complementary.

## 4.3 TRANSFER LEARNING

In computer vision, the scope of transfer learning has been enlarged to include knowledge transfer from self-supervised tasks trained on unlabeled data to supervised tasks (Doersch & Zisserman (2017)). This inspired us to reconsider our multitask learning approach from a transfer learning perspective. In this variant of transfer learning, we jointly "pre-train" our three self-supervised tasks on purely unlabeled data to convergence. We follow this up with a fine-tuning stage, using a much smaller quantity of labeled data, to train a supervised task. We carried out transfer learning experiments on the same trio of tasks tackled above in our multitask learning experiments. The results (see Table 3) favor transfer learning over simultaneously training all tasks together.

Table 3: Top-1 classification accuracy (%) with transfer learning (pre-training on self-supervised tasks followed by supervised fine-tuning) showed large improvements on all three tasks.

|          | TAG   | ID    | SC    |
| -------- | ----- | ----- | ----- |
| Transfer | 61.96 | 60.64 | 94.61 |

## 5 FUTURE DIRECTIONS

The present work developed the following theme: faced with training an audio task on limited quantities of labeled data, one can expect performance gains by jointly training the supervised task together with multiple self-supervised tasks using a WaveNet-based model operating directly on raw audio waveforms. We have shown that the improved performance on the supervised tasks scales with the quantity of unlabeled data and can be used to supplement existing data augmentation schemes.

Predicated on the performance gains observed on three fairly distinct audio classification tasks, we expect our approach to generalize to a broad range of supervised audio tasks.

Our methodology and results suggest many interesting directions for further development. Is there a limit on the number of auxiliary tasks that a single model at fixed capacity can benefit from, and can one place bounds on the expected improvement in performance? Intuitively, we expect that when our multitasking model learns to simultaneously forecast frames of audio, remove noise from the audio and perform upsampling, it must have formed a representation of the audio. What is this representation? Can it be extracted or distilled? A proper exploration of these questions should enable us to handle a broader range of auditory tasks.

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

## 6 APPENDIX

### 6.1 WAVENET TRUNK

Although audio tag classification does not require the fine temporal resolution found in raw audio waveforms, our chosen auxiliary tasks (or any arbitrary auditory task for which we may desire our model to be sufficient) require higher temporal resolutions. To satisfy this, we chose to build our model following the WaveNet architecture (van den Oord et al. (2016)).

WaveNet models are autoregressive networks capable of processing high temporal resolution raw audio signals. Models from this class are ideal in cases where the complete sequence of input samples is readily available. WaveNet models employ causal dilated convolutions to process sequential inputs in parallel, making these architectures faster to train compared to RNNs which can only be updated sequentially.

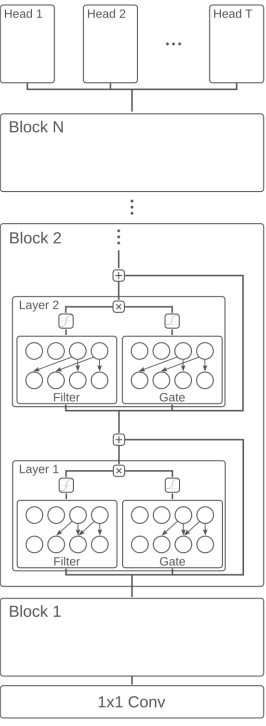

Figure 1: Multiple tasks are processed using small, task-specific neural networks built atop a task-agnostic trunk. The trunk architecture principally follows the structure of WaveNet, with several blocks of stacked, dilated, and causal convolutions between every convolution layer. Outputs from the trunk are fed into task-specific heads (details in Section 6.1).

As shown Figure 6.1, our WaveNet trunk is composed of $N$ blocks, where each block consists of $S$ dilated causal convolution layers, with dilation factors increasing from 1 to $2^S - 1$, residual connections and saturating nonlinearities. We label the blocks using $b = 1, \cdots, N$. We use indices $\ell \in [1 + (b-1)S, bS]$ to label layers in block $b$. Each layer, $\ell$, of the WaveNet trunk consists of a "residual atom" which involves two computations, labeled as "Filter" and "Gate" in the figure. Each residual atom computation produces a hidden state vector $h^{(\ell)}$ and a layer output $x^{(\ell)}$ defined via

$$
\begin{aligned}
h^{(\ell)} &= \sigma\big(W_{gate}^{(\ell)} \circledast_\ell x^{(\ell-1)}\big) \odot \tanh\big(W_{filter}^{(\ell)} \circledast_\ell x^{(\ell-1)}\big) \\
x^{(\ell)} &= x^{(\ell-1)} + h^{(\ell)}
\end{aligned}
$$

where $\odot$ denotes element-wise products, $\circledast$ represents the regular convolution operation, $\circledast_\ell$ denotes dilated convolutions with a dilation factor of $2^{\ell \bmod bS}$ if $\ell$ is a layer in block $b+1$, $\sigma$ denotes the sigmoid function and $W_{gate}^{(\ell)}$ and $W_{filter}^{(\ell)}$ are the weights for the gate and filter, respectively.

The first ($\ell = 0$) layer – represented as the initial stage marked "$1 \times 1$ Conv" in Figure 6.1 – applies causal convolutions to the raw audio waveforms $X = (X_1, X_2, \cdots, X_T)$, sampled at 16 kHz, to produce an output $x^{(0)} = W^{(0)} \circledast X$.

Given the structure of the trunk laid out above, any given block $b$ has an effective receptive field of $1 + b(2^S - 1)$. Thus the total effective receptive field of our trunk is $\tau = 1 + N(2^S - 1)$. Following an extensive hyperpameter search over various configurations, we settled on $N = 3$ blocks comprised of $S = 6$ layers each for our experiments. Thus our trunk has a total receptive field of $\tau = 190$, which corresponds to about 12 milliseconds of audio sampled at 16kHz.

### 6.2 TASK-SPECIFIC HEADS

As indicated above, each task-specific head is a simple neural network whose input data is first constrained to pass through a trunk that it shares with other tasks. Each head is free to process this input to its advantage, independent of the other heads.

Each task also specifies its own objective function, as well as a task-specific optimizer, with customized learning rates and annealing schedules, if necessary. We arbitrarily designate supervised

tasks as the primary tasks and refer to any self-supervised tasks as auxiliary tasks. In the experiments reported below, we used "audio tagging" as the primary supervised classification task and "next-step prediction", "noise reduction" and "upsampling" as auxiliary tasks training on various amounts of unlabeled data. The parameters used for each of the task specific heads can be found in Table 4 of the accompanying supplement to this paper.

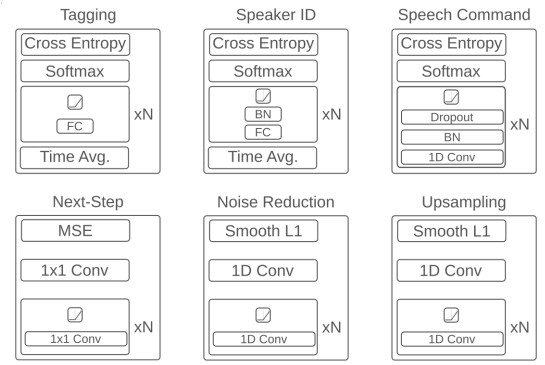

Figure 2: The head architectures were designed to be simple, using only as few layers as necessary to solve the task. Simpler head architectures force the shared trunk to learn a representation suitable for multiple audio tasks.

### 6.2.1 NEXT-STEP PREDICTION

The next-step prediction task can be succinctly formalized as follows: given a sequence $\{x_{t-\tau+1}, \cdots, x_t\}$ of frames of an audio waveform, predict the next value $x_{t+1}$ in the sequence. This prescription allows one to cheaply obtain arbitrarily large training datasets from an essentially unlimited pool of unlabeled audio data.

Our next-step prediction head is a 2-layer stack of $1 \times 1$ convolutional layers with ReLU nonlinearities in all but the last layer. The first layer contains 128 units, while the second contains a single output unit. The head takes in $\tau$ frames of data from the trunk, where $\tau$ is the trunk's effective receptive field, and produces an output which represents the model's prediction for the next frame of audio in the sequence. The next-step head treats this as a regression problem, using the mean squared error of the difference between predicted values and actual values as a loss function, i.e. given inputs $\{x_{t-\tau+1}, \cdots, x_t\}$, the head produces an output $y_t$ from which we compute a loss $\mathcal{L}_{\text{MSE}}(t) = (y_t - x_{t+1})^2$ and then aggregate over the frames to get the total loss.

We would like to note that the original WaveNet implementation treated next-step prediction as a classification problem, instead predicting the bin-index of the audio following a $\mu$-law transform. We found that treating the task as a regression problem worked better in multitask situations but make no claims on the universality of this choice.

### 6.2.2 NOISE-REDUCTION

In defining the noise reduction task, we adopt the common approach of treating noise as an additive random process on top of the true signal: if $\{x_t\}$ denotes the clean raw audio waveform, we obtain the noisy version via $\hat{x}_t := x_t + \xi_t$ where $\xi_t$ an arbitrary noise process. For the denoising task, the model is trained to predict the clean sample, $x_t$, given a window $\{\hat{x}_{t-\frac{1}{2}(\tau-1)}, \cdots, \hat{x}_{t+\frac{1}{2}(\tau-1)}\}$ of noisy samples. Formally speaking, the formulation of the next-step prediction and denoising tasks are nearly identical, so it should not be surprising to find that models with similar structures are well-adapted to solving either task. Thus, our noise reduction head has a structure similar to the next-step head. It is trained to minimize a smoothed L1 loss between the clean and noisy versions of the waveform inputs, i.e. for each frame $t$, the head produces an output $\hat{y}_t$, and we compute the loss

$$\mathcal{L}_{\text{smooth L1}}(t) = \begin{cases} \frac{1}{2}|\hat{y}_t - x_t|^2 & \text{if } |\hat{y}_t - x_t| < 1 \\ |\hat{y}_t - x_t| - \frac{1}{2} & \text{if } |\hat{y}_t - x_t| \geq 1 \end{cases} \tag{1}$$

and then aggregate over frames to obtain the total loss. We used the smooth L1 loss because it provided a more stable convergence for the denoising task than mean squared error.

### 6.2.3 UPSAMPLING

In the same spirit as the denoising task, one can easily create an unsupervised upsampling task by simply downsampling the audio source. The downsampled signal serves as input data while the original source serves as the target. Upsampling is an analog of the "super-resolution" task in computer vision.

For the upsampling task, the original audio was first downsampled to 4 kHz using the resample method in the librosa python package (McFee et al. (2017)). To keep the network operating at the same time scale for all auxiliary tasks, we repeated every time-point of the resampled signal 4 times so as to mimic the original signal's 16 kHz sample rate. The job of the network is then to infer the high frequency information lost during the transform.

Again, given the formal similarity of the upsampling task to the next-step prediction and noise-reduction tasks, we used an upsampling head with a structure virtually identical to those described above. As with the denoising task, we used a smooth L1 loss function (see eqn. (1) above) to compare the estimated upsampled audio with the original.

### 6.3 TRAINING

We trained the model using raw audio waveform inputs taken from the FSDKaggle2018 and Librispeech datasets. All code for the experiments described here was written in the PyTorch framework Paszke et al. (2017). All audio samples were first cropped to two seconds in duration and downsampled to 16 kHz. To normalize for the variation in onset times for different utterances, the 2 seconds were randomly selected from the original clip. Samples shorter than 2 seconds were zero padded. We then scaled the inputs to lie in the interval $[-1, 1]$. The noise-reduction task required noisy inputs which we obtained by adding noise sampled from ChiME3 datasets Barker et al. (2015) at a randomly chosen SNR from 10dB to 15dB. The noise types include booth (BTH), on the bus (BUS), cafe (CAF), pedestrian area (PED), and street junction (STR)) . Starting with the main task, we first performed a hyperparameter search over the number of blocks in the trunk, the number of layers per block, the number of layers and units of the main task head, and the learning rate. We tried several values for the number of blocks in the trunk, ranging from 2 to 5. We also varied the number of dilated convolution layers in each block from 3 to 8. We found that the performance and training characteristics of the network were largely unaffected by the exact architecture specifications, though learning rate was often important. We then searched over the depth and width of each auxiliary task head, as well as the learning rate for the head. These searches were done by pairing each task individually with the main task. The final choice of hyper-parameters was made by picking values which gave the best possible performance on both the main task and the auxiliary tasks, heuristically favoring performance on the main task.

We jointly trained the model on all tasks simultaneously by performing a forward pass for each task, computing the loss function for each task, and then calculating the gradients based on a weighted sum of the losses, *viz.* $\mathcal{L}_{\text{total}} = \sum_i \alpha_i \mathcal{L}_i$, where the sum runs over all the tasks. We used a uniform weighting strategy in our current experiments. More advanced weighting strategies showed no benefit for the tagging task.

We used the "Adam" optimizer Kingma & Ba (2014) with parameters $\beta_0 = 0.9$, $\beta_1 = 0.99$ , $\varepsilon = 10^{-8}$. The learning rate was decayed by a factor of .95 every 5 epochs, as this was found to improve convergence. We used a batch size of 48 across all experiments, since it was the largest batch size permissible by the computational resources available to us. Adding the noise reduction and upsampling tasks required a separate forward propagation of the noisy and downsampled audio, respectively. Exact values for all important parameters of the model can be found in Table 4.

## 6.4 HYPERPARAMETERS

Table 4: Important hyperparameter values for all experimental runs

|  | Parameter | Value |
|---|---|---|
| Trunk | # Blocks | 3 |
|  | # Layers | 6 |
|  | # Units | 64 |
| Optimizer | Type | Adam |
|  | Learning rate | $3 \times 10^{-4}$ |
|  | Epochs per step | 5 |
|  | Schedule multiplier | 0.95 |
| Audio Tagging Head | # Layers | 1 |
|  | # Units - hidden | 512 |
|  | # Units - output | 41 |
|  | Learning rate | $5.37 \times 10^{-5}$ |
|  | Epochs per step | 5 |
|  | Schedule multiplier | 0.95 |
| Next-step Head | # Layers | 2 |
|  | # Units - hidden | 128 |
|  | # Units - output | 1 |
|  | Learning rate | $5 \times 10^{-3}$ |
|  | Epochs per step | 5 |
|  | Schedule multiplier | 0.95 |
| Noise Reduction Head | # Layers | 2 |
|  | # Units - output | 128 |
|  | Filter width | 11 |
|  | Learning rate | $5 \times 10^{-3}$ |
|  | Epochs per step | 5 |
|  | Schedule multiplier | 0.95 |
| Upsampling Head | # Layers | 2 |
|  | # Units | 128 |
|  | # Units - output | 1 |
|  | Filter width | 11 |
|  | Learning rate | $5 \times 10^{-3}$ |
|  | Epochs per step | 5 |
|  | Schedule multiplier | 0.95 |

