# OpenReview forum: "LABEL-EFFICIENT AUDIO CLASSIFICATION THROUGH MULTITASK LEARNING AND SELF-SUPERVISION"
_ICLR.cc/2019/Workshop/LLD — LLD 2019_

### Official Review · AnonReviewer2 · 2019-03-31
**Accept**

**Rating:** 3
**Confidence:** 3

**Review:**

Summary:
The paper proposes a method for using multi-task learning with a variety of supervised and unsupervised audio datasets to train a convolutional network model that operates on the raw waveform. The results show that their technique improves results on three datasets compared to training on those datasets alone, and the results scale with the amount of training data.

Pros:
The paper is well written and a pleasure to read. The experiments are rigorous and well explained.

I think this is the first time I've seen a multi-task architecture for raw audio.

Cons:
These datasets are not what I'd call "limited labeled data"---Google Speech Commands, for example, has thousands of examples of each word. The datasets are smaller than LibriSpeech, which you use as the big unsupervised dataset, but ironically LibriSpeech probably counts more as "limited labeled data": consider how many times the word "xylophone" might turn up (probably not often). Perhaps a better experiment would be to recognize these rare words, given just a few examples, and see how well multi-task learning + unsupervised learning helps with that.

The paper "Listening to the World Improves Speech Command Recognition" (https://arxiv.org/abs/1710.08377v1, accepted to AAAI 2017) is about a related approach: transfer learning, as opposed to multi-task learning, for audio tasks. You should cite this paper and compare your method with theirs.

93% accuracy on Google Speech Commands seems weirdly low. I myself have trained a model that operates on the raw waveform, without any data augmentation/multi-task learning, and gets 95% accuracy for the full 30 commands, not just the 12 labels you picked. It might be because you don't use any recurrent layers and just use convolutional layers. It's not that important given the other results, but maybe it indicates a bug. I hope you release your code!

---

### Official Review · AnonReviewer1 · 2019-04-08
**Intriguing approach, but unclear relationship to transfer learning**

**Rating:** 3
**Confidence:** 2

**Review:**

The authors propose a technique for improving the performance of audio classification neural networks by simultaneously training them to solve various auxiliary self-supervised tasks. This is achieved by sharing a common "trunk" network with multiple "head" networks, each tailored to solve a specific task. Typically, a network will be optimized with four head networks, one to solve a main task and the rest to solve three auxiliary tasks. Since the trunk is shared for the different tasks, the idea is that the auxiliary tasks will result in an improved trunk network compared to only optimizing the main objective. The authors present experimental results on three main tasks: audio tagging, speaker identification, and speech command classification. These were coupled with three auxiliary, self-supervised tasks: next-step prediction, denoising, and upsampling. Since training data for these auxiliary tasks could be created from unlabeled data, it represents a virtually inexhaustible supply of potential training data. Experimental results for this architecture showed a significant improvement was obtained by joint training with these auxiliary tasks. However, the authors also present results for a transfer learning experiment, where the network was first trained on the auxiliary tasks and then fine-tuned for the main task. This yielded an even greater improvement, which calls into question the utility of the proposed method. Unfortunately, no analysis of explanation of this fact is presented.

The description of the method is quite comprehensive, but at times vague and lacking some details. First, while the paper describes three main tasks (audio tagging, speaker identification, and speech command classification), much of the description of the architecture only refers to audio tagging. The method for the joint training is also only mentioned towards the end, in the appendix. It would be clearer if some of these details were included in the main text. It is also not completely clear whether the trunk network is jointly trained between all three main tasks and the auxiliary tasks, or between a main task and the auxiliary tasks. While it is likely the latter, it is never clearly spelled out. There is also little motivation for the auxiliary tasks. Why were these chosen as opposed to others? For example, what do the authors expect the next-step prediction task to add? If it is simply a matter of predicting the next sample in a time series, this shouldn't necessary require a very high-level knowledge of the audio signal structure, since continuity already provides a very strong prior. Similar questions can be posed for the other auxiliary tasks. Furthermore, some auxiliary tasks are trained with an L^2 loss, while others are trained with a smoothed L^1 loss. Why were these different choices made? Finally, the method of choosing hyperparameters is also not clear. The authors state that they were chosen "heuristically favoring performance on the main task". What does this mean?

The experiments provide interesting results, but are not as complete as they could be. For example, there are several popular datasets for audio tagging. why were the particular datasets chosen? More importantly, why are no results for state-of-the-art methods presented? It is not necessary that the proposed architecture perform better than the current state of the art, but it is important to provide a context for the results. The authors also add a significant amount of training data to the problem through the auxiliary training tasks, but neglect to discuss any impact on training time. While this may not always be an issue, it is a relevant trade-off to be made when considering when to adopt the proposed architecture. Finally, the authors make a claim about additivity (or complementarity) of their proposed approach when coupled with data augmentation. It would be interesting to see to what extent the changed labels overlap between the two methods. If they are indeed complementary, there would be little overlap between the set of labels changed by adding one and then the other.

Finally, the transfer learning results (as mentioned above), pose a significant problem with respect to the utility of the proposed algorithm. If we can simply train the network trunk for the auxiliary tasks separately, and then add a new head network and train that for the main task, why should we consider the whole apparatus proposed in the current manuscript? This holds doubly true considering that the transfer learning approach significantly outperforms the proposed approach for the considered tasks. There may be some properties like reduced training time, fewer hyperparameters to select, and so on, but no discussion is provided. Since this is a potentially very important alternative, analysis and discussion of the merits of the two approaches is strictly necessary.

Some more minor comments follow:
- With respect to general-purpose audio representations, the authors may want to mention the audio scattering transforms developed by Mallat and collaborators.
- In Section 3, the authors mentioned experiments being made with "0 to 3" auxiliary tasks. The subsequent experiments only present results for 0 _and_ 3 auxiliary tasks, with no results for 1 or 2 tasks. This should be corrected.
- One dataset is described as containing "uncompressed PCM", another as "WAVE format files", while the format of the third is not specified. If the authors insist on including this information, they should be consistent in their descriptions. What is the format of the third dataset? Are the WAVE files also stored as PCM? Or are they stored in µ-law or some other format?
- The authors make the claim that "spectral/cepstral representations of audio ... significantly restrict the range of audio processing tasks which they can perform". A finely sampled spectral representation contains enough information for synthesizing a new signal which sounds virtually identical to the original. Where they may fall short is in sample-by-sample reconstructions, since they do not include the phase. One could argue that this sample-by-sample reconstruction is rarely what's desired in audio classification tasks. Indeed, the type of tasks for which they are necessary mostly includes low-level processing tasks such as the auxiliary tasks introduced in this paper. The fact that that there is such an "impedance mismatch" between the main and auxiliary tasks should be cause for concern.
- The MAP@3, Top-1, and Top-5 metrics, although well-known, should be defined completeness.
- The difference between the "baseline" and "none (0)" rows in Table 1 is a bit subtle. While the second does not include any unlabeled data, it still performs joint training on the main and auxiliary tasks but on the main task's training data. This is not obvious from a first glance and should be clarified.
- In Table 2, it would be useful to provide results for "NI + PS", "NI + PS + MTL100" and the same for "MTL500". In the interest of space, however, it may be better to simply sketch these results in the text if they do not add too much more information.
- The authors state that "Interestingly, the performance gains from augmenting with noisy data are similar to those obtaining by training the main task jointly with a self-supervised noise-reduction task." Why is this interesting? Why could this be the case? Is there a similarity in label assignment as well? I suggest the authors finish this train of though.
- In Table 3, please write out the full names of the tasks as in Table 1.
- There are several capitalization errors in the bibliography. In particular, several uppercase characters have been converted to lowercase: "English", "Mandarin", "ChiME", "PyTorch".
- Having figures in gray make them hard to read, especially when printed. Unless there is some compelling reason not to, I suggest they be regenerated in black.
- Please provide a definition for dilated convolution.
- The sequence in Section 6.2.1 should be delimited by parentheses, not curly braces (which delimit sets).
- The "smoothed L^1" norm is also known as the Huber loss in statistics and other fields.
- "Python" has the first letter capitalized.
- Please define SNR.

Finally, I strongly suggest the authors make their code available to the general public.

---

### Decision · Program_Chairs · 2019-04-16
**Acceptance Decision**

Accept